# Novel Calcitonin Gene-Related Peptide (CGRP) Interfering Migraine Therapies and Stroke—A Review

**DOI:** 10.3390/ijms252111685

**Published:** 2024-10-30

**Authors:** Michael Thomas Eller, Florian Frank, Katharina Kaltseis, Anel Karisik, Michael Knoflach, Gregor Broessner

**Affiliations:** 1Department of Neurology, Medical University of Innsbruck, 6020 Innsbruck, Austria; michael.eller@i-med.ac.at (M.T.E.);; 2VASCage—Centre on Clinical Stroke Research, 6020 Innsbruck, Austria

**Keywords:** calcitonin gene-related peptide, CGRP, stroke, migraine, gepants, monoclonal antibodies

## Abstract

Migraine and stroke are neurological disorders with significant global prevalence and impact. Recent advances in migraine therapy have focused on the calcitonin gene-related peptide (CGRP) pathway. This review examines the shared pathomechanisms between migraine and stroke, with emphasis on the role of CGRP. We analyze the current literature on CGRP’s functions in cerebrovascular regulation, edema formation, neuroinflammation, and neuroprotection. CGRP acts as a potent vasodilator and plays a crucial role in trigeminovascular activation during migraine attacks. In stroke, CGRP has demonstrated neuroprotective effects by improving collateral circulation and reducing ischemia-reperfusion injury. Concerns have been raised about the potential impact of CGRP inhibitors on stroke risk and outcomes. Studies in animals suggest that CGRP receptor antagonists may worsen cerebral ischemia by impairing collateral flow. We discuss the implications of these findings for the use of CGRP-targeting therapies in migraine patients, especially those at increased risk of stroke. Additionally, we explore the complex interplay between CGRP, endothelial function, and platelet activity in both conditions. This review highlights the need for further research to elucidate the long-term cerebrovascular safety of CGRP pathway inhibitors and to identify potential subgroups of migraine patients who may be at higher risk of adverse cerebrovascular events with these novel therapies.

## 1. Physiological Effects of Calcitonin Gene-Related Peptide (CGRP)

The 37 amino acids that comprise CGRP, along with its two subtypes, alpha- and beta-CGRP, are mainly expressed in neurons of the central and peripheral nervous system. A high density of CGRP receptors has been reported in the trigeminal nerve axis, particularly in the trigeminal ganglion (TG). CGRP has also been suggested to occur in non-neuronal cells such as lymphocytes, mononuclear cells, adipocytes, and endothelial and epithelial cells [1,2,3]. Ischemia-induced TG activation leads to CGRP release in the meningeal vasculature. CGRP binds to CLR:RAMP1 (calcitonin receptor-like receptor, receptor activity modifying protein 1), a member of the CLR:RAMP family. The G protein-coupled CLR must heterodimerize with one of the three RAMPs 1–3 to become biologically active. Subsequent receptor downstream results in cAMP accumulation, calcium mobilization, and nitric oxide (NO) release, resulting in a following vasodilation [2,4]. Besides CGRP’s vasoactive potential, it sensitizes the NMDA (N-Methyl-D-Aspartate) receptor and activates direct and indirect anti-apoptotic and growth-factor signaling [5]. These findings propose CGRP as a crucial player in collateral blood flow during the acute phase of ischemic stroke and as a potential neuroprotectant in subarachnoid and intracerebral hemorrhage [6,7]. Additionally, CGRP has been found to upregulate Bcl2/Bax (B-cell lymphoma 2/Bcl-2-associated X protein), indicating mitochondrial stabilization and the prevention of apoptosis [8]. Recently, CGRP has been suggested to promote edema formation, tumor necrosis factor alpha elevation, and accumulation of neutrophils in mouse skin [9,10].

In the nociceptive system, CGRP represents one of the most involved neuropeptides. The nociceptive potential is mediated by the activation of the CLR:RAMP1 complex, which subsequently sensitizes cells [3,11]. It is distributed widely across the nociceptive axis, playing a significant role, particularly in those who experience migraine. With a lower affinity, CGRP also binds to CTR:RAMP1 (calcitonin receptor), a receptor complex in which amylin primarily acts as a ligand. CGRP is not the sole ligand for the CLR:RAMP family. Adrenomedullin (AM) has been observed to exhibit a preferential binding affinity for the CLR:RAMP2 (AM1), followed by the CLR:RAMP3 (AM2). Furthermore, evidence indicates that CLR:RAMP2/3 receptors possess substantial vasodilative, cardioprotective, and anti-apoptotic effects, whereas CTR:RAMP1 has not been demonstrated to exhibit such functions. Amylin (and CGRP) predominantly mediate an increase in gastrointestinal (GI) motility via CLR:RAMP1, which could potentially lead to a CGRP-mediated GI hyperactivity, as observed in some migraine patients [2,12].

Migraine is a very common neurovascular disorder that affects about 15% of the general population, predominantly women, and is the third leading cause of global years lived with disability [13]. CGRP levels are elevated in the jugular vein during an acute migraine attack [14]. Moreover, CGRP elevation has been found in tears and saliva during and between acute migraine attacks [15]. In chronic migraine patients, an elevation of plasma CGRP levels has been reported compared to healthy control groups. An intravenous (i.v.) infusion of CGRP in migraine patients compared to non-migraine patients induced a migraine-like headache in the migraine patients’ group, whereas in the control group, CGRP infusion induced only a mild headache without migraine-like symptoms, suggesting that migraine patients are more sensitive to CGRP [16,17,18]. The final and possibly most convincing evidence of CGRP’s profound involvement in the pathogenesis of migraine is the effectiveness of CGRP targeting therapies in migraine prevention and acute treatment.

## 2. Migraine Treatment with CGRP-Pathway Targeting Therapeutics

CGRP has now become the focus of numerous studies demonstrating a significant impact in the prevention and abortion of migraine attacks via monoclonal antibodies or gepants. Four monoclonal antibodies (Erenumab, Galcanezumab, Fremanezumab, and Eptinezumab) are currently on the market and have been approved for episodic migraine and chronic migraine. They are applied subcutaneously (Erenumab, Galcanezumab, and Fremanezumab) or intravenously (Eptinezumab). Due to their long half-life of 21–50 days, the frequency of administration varies between monthly for Erenumab, Galcanezumab, and Fremanezumab and every three months for Eptinezumab and high-dose Fremanezumab [19]. Erenumab binds to the canonical CGRP receptor CLR:RAMP1 at its extracellular domain. With lower affinity, it interacts with CLR:RAMP2, CLR:RAMP3, and CTR:RAMP1, physiologically activated by adrenomedullin (CLR:RAMP1 and CLR:RAMP2) and amylin (CTR:RAMP1). By binding at CTR:RAMP1, Erenumab could lead to decreased GI activity and the commonly reported side-effect of constipation [12]. Eptinezumab, Fremanezumab, and Galcanezumab bind to the receptor-binding site of endogenous alpha- and beta-CGRP, with no evidence of additional interactions (see Figure 1) [20]. Due to their high molecular weight of around 150,000 Da, mAbs are unable to cross the intact blood-brain barrier (BBB) in significant amounts. However, one study has raised data that mAbs may act within the vessel wall or indirectly influence cerebral vasomotor potential [21]. The primary site of action for mAbs is considered to be the trigeminal ganglion and the dura mater, characterized by a high density of CGRP receptors and a location outside of the BBB. Fluorescently labeled Fremanezumab has not been observed to cross the intact BBB in rats. However, high amounts were detected in dural vessels, the dura mater, the trigeminal ganglion, the C2 dorsal nerve ganglion, the sphenopalatine ganglion, and the superior sympathetic cervical ganglion. No signals of fluorescently labeled Fremanezumab were detected in the cortex, thalamus, hypothalamus, or spinal trigeminal nucleus [1,22,23].

With a mean molecular weight of 600 Da, gepants are of significantly smaller molecular weight than mAbs. This might implicate a simplified penetration of the BBB for gepants. However, only small amounts of MK-3207, a studied CGRP receptor antagonist, have been observed to cross the BBB. For Olcegepant, no central nervous point of action could be shown [24,25].

Gepants are the newest treatments approved for migraine headaches in the United States and some European countries. Ubrogepant is approved for acute migraine treatment, while Atogepant is approved for prophylactic migraine treatment in episodic migraine patients. Rimegepant is approved for both acute and prophylactic use in episodic migraine patients [19,26]. Gepants are small molecules that exhibit a high affinity for the RAMP1 subunit of the canonical CGRP receptor, with an almost insignificant affinity for the AM receptors CLR:RAMP2 and CLR:RAMP3. Due to their selectivity for the RAMP1 subunit, gepants can also interact with the amylin receptor CTR:RAMP1. In contrast to Erenumab, constipation is not a frequently reported adverse effect in patients undergoing therapy with gepants [20]. Unlike subcutaneously applied mAbs, gepants are available as oral treatments. Ubrogepant is a medication for acute migraine that should be taken in a manner like triptans for treating acute migraine headaches. Atogepant and Rimegepant are both used for the prevention of migraines. The frequency of use for Atogepant is daily, while for Rimegepant, every other day, and Rimegepant may also be used as acute treatment on days when no prophylactic therapy tablet is taken. Zavegepant is another FDA-approved gepant for acute migraine treatment administered by using a nasal spray [27]. Figure 1 illustrates the targets of all currently approved CGRP pathways targeting migraine drugs.

## 3. The Complex Relationship Between Migraine and Stroke: Insights into Risk Factors and Vascular Mechanisms

The relationship between migraine and stroke is well covered in other reviews [28,29,30,31]; therefore, we will focus on the most important points. In a large epidemiological study, migraine with aura (MA) was among the four most important risk factors for stroke and is more strongly associated with stroke and other cardiovascular diseases (CVD) than obesity or unfavorable lipid levels. However, the absolute risk of stroke in individuals aged < 45 years is modest [30]. However, the stroke risk within migraineurs with aura is higher for women, women who smoke, and women who take combined oral contraceptives. Therefore, it is recommended that women with MA avoid taking combined oral contraceptives, which may limit their contraceptive options [31]. Observational studies have suggested a possible association between patent foramen ovale (PFO) and migraines, as well as an increased incidence of PFO in patients with MA, resulting in a higher risk of paradoxical embolism [32].

In a Danish population-based, nationwide study, a connection between migraine with aura and cardioembolic stroke due to atrial fibrillation (AF) has been reported [33]. However, no correlation between MA and AF has been found in an Italian cohort study, suggesting a complex condition of stroke etiology in migraineurs [34]. Particularly in the male population, an increased risk of carotid artery dissection has been observed in migraine without aura (MO). Additional data also suggests an elevated risk for hemorrhagic stroke in migraineurs [30,35].

Increasing evidence suggests that migraine is a predictor of the risk of non-stroke vascular events, such as myocardial infarction, as well as a potent risk factor for any vascular event per se [30].

In support of these findings, studies have identified endothelial dysfunction in the migraine population. They have described elevated numbers of endothelial progenitor cells and endothelial microparticles in migraineurs, indicating a disrupted endothelium [36,37,38]. Moreover, migraineurs exhibited increased platelet adhesion and activation, especially female migraineurs with aura, showed elevated levels of fibrinogen and factor II, suggesting that these patients may be in a hypercoagulable state with altered platelet activity [39,40,41,42,43]. Although these findings are limited by inconsistent study designs and heterogeneous inclusion and exclusion criteria, one can assume that some endothelial alterations are present in migraineurs. However, it is difficult to conclude whether these endothelial alterations cause migraines or if they are a result of this complex neurological disorder. C-reactive protein (CRP), a marker of inflammation and vascular disease, is elevated in individuals who suffer from migraine. CRP levels were found to be elevated in patients with both MA and MO, especially in women between the ages of 19 and 34 with MO, compared to healthy women of the same age [44,45,46,47]. Other pro-inflammatory cytokines, IL-1 and IL-6, are also elevated in migraineurs, supporting the theory of a (pro-)inflammatory state in this population [39,48]. In support of these findings, studies have identified endothelial dysfunction in the migraine population and increased platelet adhesion and activation [37,38,39].

## 4. The Role of Attack Medication and Cortical Spreading Depression (CSD)

Triptans and ergot alkaloids are two drugs commonly used to treat acute migraines. Triptans selectively bind to the 5-HT1B, 5-HT1D, and 5-HT1F receptors, while ergot alkaloids have a wider range of receptor targets, resulting in a less favorable adverse effect profile compared to triptans. By binding to the 5-HT1B/1D receptors, triptans and ergots have a vasoconstrictive potential. It has been suggested that the excessive use of ergotamine may be associated with an increased risk of cerebrovascular and cardiovascular events. For triptans, this association has not been found. Nevertheless, it is advisable to exercise caution when prescribing triptans to patients with a history of cardiovascular or cerebrovascular disease [49,50].

Cortical spreading depression is a phenomenon characterized by a wave of depolarization that starts in the occipital lobe and migrates to the frontal lobe at a rate of approximately 1–2 mm/s. There is considerable evidence that CGRP plays a complex role in the development of CSD, primarily through its effects on cerebral blood vessels and the trigeminovascular system. However, CGRP-pathway mAbs have not been shown to reduce migraine aura in patients with MA [51,52]. Cortical spreading depression is associated with the clinical manifestation of a migraine aura, whereas CSD has also been observed in the penumbral tissue of ischemic stroke patients. CSD causes an ion translocation between neuronal cell bodies and the interstitium and an increase in energy consumption. In healthy brain tissue, increased energy turnover can be balanced by subsequent vasodilation. However, in energy-compromised tissue, such as the penumbra in ischemic stroke, this CSD could result in cell death. CSD has also been observed to result in the release of excitotoxic glutamate, which could lead to an additional risk of cell death in the penumbra [53]. Figure 2 identifies the most common and significant risk factors for stroke in migraineurs.

## 5. Molecular Implications of CGRP’s Role in Stroke

In acute ischemic stroke, the BBB becomes compromised, leading to increased permeability for substances normally restricted from entering the brain. A compromised BBB could also result in the formation of cerebral edema. CGRP has been implicated in promoting the development of edema, although conflicting findings suggest it may also play a protective role by preventing BBB injury and reducing infarct size [10,54,55]. Even if these results are conflicting, the beneficial effects of CGRP in ischemic events are far more prominent. CGRP and AM exhibit significant vasodilatory potential in the microvasculature [56]. During sudden drops in systemic blood pressure, CGRP helps maintain cerebral blood flow (CBF) [57]. However, reintroducing blood flow after ischemic conditions can lead to reperfusion injury due to oxidative stress, and CGRP offers both vasodilatory and antioxidative properties [58]. Additionally, pharmacologic preconditioning with morphine has been shown to stimulate systemic CGRP release, which reduces reperfusion injuries, indicating a potential interaction with the opioid signaling pathway [59]. This protective effect of CGRP can be diminished by triptans, which reduce CGRP release. CGRP demonstrates significant anti-inflammatory effects, particularly relevant in neuroprotection during stroke and other inflammatory conditions. CGRP inhibits the release of pro-inflammatory cytokines such as TNF-alpha, IL-1β, and IL-6 from various immune cells, effectively reducing the inflammatory response that can exacerbate tissue damage during and after a stroke [9,60].

CGRP’s neuroprotective anti-apoptotic effects were found in rodents mimicking ischemic events. It turned out that these effects are primarily mediated through the CREB (cyclic AMP response element-binding transcription factor) pathway, which upregulates Bcl-2 and subsequently reduces caspase-3 levels [61,62]. Caspase-3 is a critical pro-apoptotic molecule, while Bcl-2 stabilizes the mitochondrial membrane through the Bcl-2/BAX pathway, also influenced by CGRP. Thus, CGRP can enhance neuronal function and reduce apoptosis in brain tissues following ischemic injury by modulating MAPK (mitogen-activated protein kinase) pathways [8,63].

In the context of hemorrhagic stroke, CGRP has been found to decrease neuronal apoptosis, reduce the risk of vasospasm, and support nerve regeneration, respectively [64]. Not only has CGRP shown beneficial effects in stroke, but AM has demonstrated anti-inflammatory effects, including a reduction in oxidant production in the cerebellum [63,64]. Table 1 lists CGRP’s potential roles and the related effects in stroke.

In brief, CGRP plays a complex and predominantly beneficial role in stroke, exhibiting vasodilatory, neuroprotective, and anti-inflammatory effects. While its impact on cerebral edema remains controversial, CGRP helps maintain cerebral blood flow, reduces reperfusion injury, inhibits pro-inflammatory cytokine release, and demonstrates anti-apoptotic properties through various molecular pathways, suggesting its potential importance in stroke management and therapy.

## 6. Cardiovascular Risk Signals in Clinical Trials Using Monoclonal Antibodies and Gepants

In a trial involving 60 patients with MO who received 70 mg of Erenumab subcutaneously every four weeks, vasomotor reactivity (VMR) and brachial flow-mediated dilation (FMD) were monitored during follow-up. The results showed no differences in cerebral or systemic vasomotor activity compared to the healthy control group [65]. However, there was little effect on cerebral hemodynamics for CGRP-targeted mAbs, especially in migraineurs responding to mAb preventive treatment. The decrease in cerebral blood flow velocity (v) in this subgroup should be critically interpreted, as the number of participants was small [6]. A meta-analysis of 3300 participants of five placebo-controlled RCTs of Erenumab raised no concerns regarding the CV risk profile [66]. However, all analyzed studies recruited subjects aged ≤ 65 years. Of all participants, 143 were aged ≥ 60 years, and 63 were enrolled in the treatment arm, representing less than 10% of all participants. It is therefore of utmost importance to acknowledge that all studies finally leading to market authorization used strict inclusion and exclusion criteria, mostly enrolling patients lacking CVD or other relevant preconditions. A post-hoc pooled analysis of long-term clinical trial data on Erenumab examined adverse events in relation to participants’ estimated 10-year CV risk. Across all risk categories (no risk, low, moderate, and high), Erenumab exposure did not lead to any significant AE signals [67].

For Eptinezumab, Fremanezumab, and Galcanezumab, RCTs included patients up to 70–75 years old with non-concerning CV safety data [19,68]. However, all the abovementioned RCTs excluded patients with pre-existing CVD, and the number of participants ≥ 60 years old was low. This highlights once more the limited evidence of RCTs regarding efficacy and safety in elderly patients, especially in those with preexisting CVDs.

A Spanish multicenter observational study enrolled patients aged 65 to 87 years, evaluating the efficacy and safety of Erenumab, Fremanezumab, and Galcanezumab [69]. Participants had a high prevalence of comorbidities, including dyslipidemia, hypertension, anxiety, depression, and ischemic cardiovascular disease. In this real-world observational study, elderly migraineurs showed a similar benefit in migraine prevention compared to younger migraineurs. No signals for CVD adverse events were raised in this elderly cohort with pre-existing comorbidities.

CVD safety studies for mAbs are not only limited by their preselected patient cohort but also by a lack of long-term safety data. However, post-marketing pharmacovigilance is common in many countries, and so far, no CVD signals have been documented. Additionally, a 5-year open-label follow-up of Erenumab has shown no CV safety concerns [70].

As gepants are the most recently approved CGRP receptor inhibitors in the treatment of migraine, no long-term CV safety data are available. Nevertheless, all phase 3 clinical trials for Atogepant, Rimegepant, and Ubrogepant showed no increase in CVD adverse events in treated patients [19,27,68]. A recent network meta-analysis found no CV concerns for Atogepant and Rimegepant in phase 2/3 studies [19]. These studies, however, excluded or underrepresented subjects > 65 years of age and patients with a history of CVD. A recently conducted controlled, open-label industry-sponsored study included 735 participants with CV risk factors to evaluate the safety of Rimegepant for acute migraine treatment with doses up to once daily [71]. The follow-up was one year, and no CV concerns were raised. However, patients with former CV events and uncontrolled CVD were excluded

## 7. Cardiovascular Risk Signals in Non-Clinical Studies

A study by Inge A. Mulder and colleagues demonstrated a significant effect of gepants on cerebral infarct volume and functional outcome after transient ischemic attack (TIA) or stroke-mimicking experiments [72]. Two gepants, Olcegepant and Rimegepant, were administered as single or repeated doses. Mice were randomized into two arms: the treatment arm or a matching placebo control arm.

To simulate a TIA, the MCA was occluded for 12 min. Prior to MCA occlusion, a single dose of Olcegepant (1 mg/kg, n = 19) or placebo (n = 18) was administered. In the Olcegepant-pretreated group, 14 out of 19 mice developed infarction, whereas 6 out of 18 mice in the naive group developed infarction.

In the second experiment, Olcegepant (1 mg/kg) or vehicle was administered before MCA occlusion for 60 min. After 60 min of occlusion, all mice developed infarction. In Olcegepant-pretreated mice, the infarct volume was more than twice as large as in Olcegepant-naive mice.

Rimegepant was further evaluated. Prior to 60 min of MCA occlusion, 100 mg/kg of Rimegepant or placebo was administered. In the Rimegepant group, 6/8 mice died compared to 0/8 in the vehicle group.

In the above-mentioned TIA model, at a dose of 10 mg/kg Rimegepant versus placebo, all mice developed infarction. In the Rimegepant-pretreated group, the infarct size was 60% larger than in the vehicle group. It should be noted that the doses of Rimegepant studied, 10 mg/kg and 100 mg/kg, are 10–100 times higher than the approved human dosage.

In the fourth set of experiments, mice were treated with 0.1 mg or 1 mg/kg of Olcegepant for two weeks. Both dose regimens increased infarct volume and neurological deficits comparable to those of a single dose of Olcegepant.

Nevertheless, other animal models investigating gepants have demonstrated that short-term inhibition of CGRP signaling does not significantly alter hemodynamic parameters, including heart rate, blood pressure, cardiac output, or coronary flow. Furthermore, no exacerbation in ischemic severity has been observed. These findings suggest that the acute blockade of CGRP-mediated pathways exerts minimal influence on systemic cardiovascular dynamics, which is in accordance with the results of the major clinical RCTs [73].

In a study by L. Ohlsson et al., CGRP receptors were blocked with Erenumab in isolated human cranial arteries, demonstrating no significant effect on vasoconstriction or vasodilation [74].

## 8. CGRP Levels in Acute Stroke Patients

Bhar-Hosseini et al. [75] evaluated CGRP levels in acute stroke patients. Blood samples were drawn from a peripheral vein at admission and again the following morning. The study cohort had a mean age of 78 (70–86) years, with 37% women and a mean NIHSS of 13 (7–18) on the day of admission. They observed normally distributed CGRP levels with no changes from admission to the morning after admission. A multimodal CT was performed at baseline to evaluate leptomeningeal collaterals, penumbral size, and the ischemic core. Final infarct volumes were measured by CT or MRI within 24 h after admission. The study found no correlation between admission or next-day CGRP levels and collateral flow, penumbral size, or infarct size. The 90-day modified Rankin Scale (mRS) was evaluated to determine any correlation with admission or next-day CGRP levels, but no such correlation was found. It is worth noting that data on CGRP in acute stroke patients are scarce, and this study is the only prospective study to measure CGRP levels in such patients.

## 9. CGRP and Other Vascular Disease

CGRP is one of the most potent vasodilators in humans, primarily acting on the microvascular bed. CGRP does not appear to play a key role in physiological blood pressure regulation, as evidenced by studies in which CGRP receptors were blocked in healthy volunteers or CGRP knockout mice [76]. However, under pathophysiologic conditions, CGRP has been found to influence arterial hypertension [77]. Some studies have reported elevated systolic and diastolic blood pressure in migraine patients treated with mAbs, while other phase 2/3 and post-marketing studies have not found such an association [78,79].

In patients with stable angina pectoris, intra-coronary injection of CGRP has been shown to delay the onset of myocardial ischemia [80]. In mice with acute myocardial infarction (MI), CGRP has been reported to significantly reduce ischemia/reperfusion injury [81]. In congestive heart failure, i.v. α-CGRP infusion improved functional cardiac parameters [82].

In mice, the administration of exogenous CGRP has been demonstrated to have antiplatelet effects by inhibiting platelet-derived tissue factor [83].

## 10. Discussion

CGRP plays a crucial role in (cranial) vasodilation. Blocking CGRP receptors with gepants can worsen ischemic stroke in mice. In contrast, none of the large, placebo-controlled, phase 2/3 trials of mAbs or gepants have raised concerns about vascular safety. These findings suggest that there may be compensatory mechanisms in migraineurs without cardiovascular diseases when the CGRP pathway is blocked. Whether this compensation is triggered by other peptides, such as adrenomedullin, remains unknown. It is important to note that these RCTs, with some exceptions, lasted no longer than 12 weeks to 6 months and mainly excluded patients with a history of CVD.

The use of mAbs and gepants as preventive treatments for migraine has been increasing in recent years, with the possibility that they may become the first-line preventive treatment for migraine in the future.

Migraine is a neurological disorder that imposes a significant burden on patients. Therefore, prophylactic treatment with targeted therapies is highly justified. Traditional oral preventive treatments have been associated with a higher number of adverse events, and some are contraindicated in women of childbearing potential, who represent the majority of migraineurs. CGRP-interfering drugs have demonstrated fewer adverse events with higher efficacy, making them an excellent option for many patients. Women < 45 with MA have a higher risk of ischemic stroke than men with MA, a risk that further increases if they smoke or use oral contraceptives. Although the role of CGRP in migraine aura pathogenesis is not yet fully understood, there is no evidence indicating that women with MA on CGRP-targeted therapies face additional stroke risk. However, patients with cerebral autosomal dominant arteriopathy with subcortical infarcts and leukoencephalopathy (CADASIL) should avoid CGRP-targeted treatment, as CGRP primarily acts on small blood vessels [84]. Yet, the predominant age of patients receiving these targeted migraine prophylactic treatments is ≤65 years. As the population of migraineurs continues to age, more patients on CGRP-interfering therapies may face stroke, MI, and other cardiovascular diseases.

A multicenter observational study was conducted on patients aged ≥ 65 years with cardiovascular comorbidities, which found no safety issues for preventive migraine treatment with mAbs [69]. However, the impact of a (long-term) blocked CGRP pathway in combination with coincident acute ischemic stroke and other CVDs remains unclear.

### 10.1. Concerns in Acute Vascular Events

Monoclonal antibodies and gepants cannot cross the BBB in significant amounts in individuals with an intact BBB. In contrast, one study has shown an alteration of cerebrovascular vasodilation by mAbs in rats [21]. In cases of acute ischemic stroke or subarachnoid hemorrhage (SAH), where the BBB is compromised, mAbs/gepants may diffuse into areas that are normally protected by the BBB. In ischemic stroke, vasodilation plays a crucial role in maintaining sufficient blood supply to the vulnerable penumbra [53]. Conversely, the CGRP pathway may be a potential target for preventing post-ischemic edema. However, the evidence supporting this hypothesis is inconclusive, with conflicting results reported in the literature. Additionally, the only study to investigate the role of CGRP in edema utilized a murine model of skin injury [10]. In acute stroke patients, CGRP levels at admission and the morning after admission did not correlate with infarct size or functional outcome after 90 days following ischemic stroke [75]. Notably, since plasma was collected from a peripheral vein, no conclusions can be drawn regarding CGRP levels in brain-supplying vessels. Obtaining comparable CGRP levels is not a trivial task, given the existing data on the half-life of CGRP of approximately 7 min [85]. A precise description of pre-analytic processing is crucial, calling for future studies investigating the role of CGRP in various vascular diseases.

### 10.2. Dual CGRP-Targeted Treatment by mAbs and Gepants

An increasing number of patients are receiving preventive migraine treatment with mAbs and are using gepants as acute attack therapy. However, there are currently only very limited safety or efficacy studies available investigating this combination. It is, therefore, essential to determine whether these patients face an increased risk of stroke per se or a worse prognosis in the event of an acute stroke.

As mentioned above, Erenumab binds with a high affinity to CLR:RAMP1 and a lower affinity to CLR:RAMP2-3. Eptinezumab, Fremanezumab, and Galcanezumab target the CGRP ligand. Gepants are selective for the RAMP1 subunit of CLR:RAMP1 but may also act at CTR:RAMP1 (amylin receptor). A randomized, open-label study evaluated the safety profile of dual CGRP-targeting therapy with the preventive use of Erenumab or Galcanezumab and Ubrogepant for acute treatment [86]. No safety issues or pharmacokinetic interactions were reported after 45 days of follow-up. Real-world data also suggest a favorable risk profile of dual CGRP-targeted therapy [87].

However, in human arteries, the combination of Erenumab with Rimegepant or Olcegepant increases the threshold at which CGRP induces vasodilation. In contrast, isolated higher doses of Erenumab alone do not show such an effect [88]. This could potentially endanger penumbral tissue in acute stroke patients on dual therapy with a compromised BBB, as both monoclonal mAbs and gepants may directly affect the microvasculature or brain tissue. The vasodilative and anti-apoptotic effects of CGRP could be inhibited by these drugs [53]. Unlike CGRP, other peptides that bind to these receptors, such as AM and amylin, are not blocked by gepants, which could allow for compensatory mechanisms in acute stroke settings [88]. Since gepants for acute treatment are infrequently used and have a shorter half-life compared to monoclonal mAbs, this may not pose a significant concern in clinical practice when treating acute migraines in migraineurs with absent CVD.

### 10.3. Molecular Findings Leading to Clinical Implications

Based on the abovementioned involvement of CGRP in stroke and other acute CV events, as well as limited data on CGRP antagonism and acute stroke, some suggestions can be made for clinical practice:

In patients > 65 years of age and relevant CVD, the individual benefits and harms of a newly initiated migraine treatment specifically acting within the CGRP pathway should be critically reconsidered. In these cases, beta-blockers or candesartan, which offer more favorable cardiovascular risk profiles, should be considered as alternative treatments. If CGRP-targeted therapy is necessary, gepants, which have a shorter half-life compared to mAbs, may be a better choice for managing potential side effects.

In patients with coincident acute stroke currently under CGRP-interfering migraine preventive treatment, these migraine treatments should be stopped immediately. Due to the long half-life of mAbs, short-term recovery could be jeopardized. Motor recovery after a stroke generally occurs within the first three months, while other remodeling processes can continue for up to one year [89,90].

Thus, reinitiating CGRP-targeted therapy after a stroke should be carefully considered on an individual basis. It is important to take stroke etiology into account, as CGRP plays a significant role in the microvasculature. Therefore, patients with microvascular disease should avoid these treatments [76]. For other post-stroke patients, gepants may be a safer alternative compared to mAbs, given their shorter half-life.

These recommendations emphasize a personalized approach to managing migraine for patients with elevated cardiovascular risk or a history of stroke, aiming to achieve a balance between treatment efficacy and patient safety.

## 11. Conclusions

In summary, while CGRP plays a crucial role in cranial vasodilation, studies have shown conflicting effects of CGRP-targeting therapies on ischemic stroke development in mice versus humans. Large clinical trials have not raised concerns about vascular safety, suggesting potential compensatory mechanisms in migraine sufferers without cardiovascular disease. However, the impact of CGRP pathway blockage on the outcome of coincidental acute stroke or other vascular events remains unclear.

Current data suggest that a cautious approach to CGRP-targeted therapies is necessary, particularly in patients with existing cardiovascular comorbidities. Further research is imperative to guide clinical decision-making and ensure patient safety. This research must include real-world studies, especially in older populations with cardiovascular comorbidities.

## Figures and Tables

**Figure 1 ijms-25-11685-f001:**
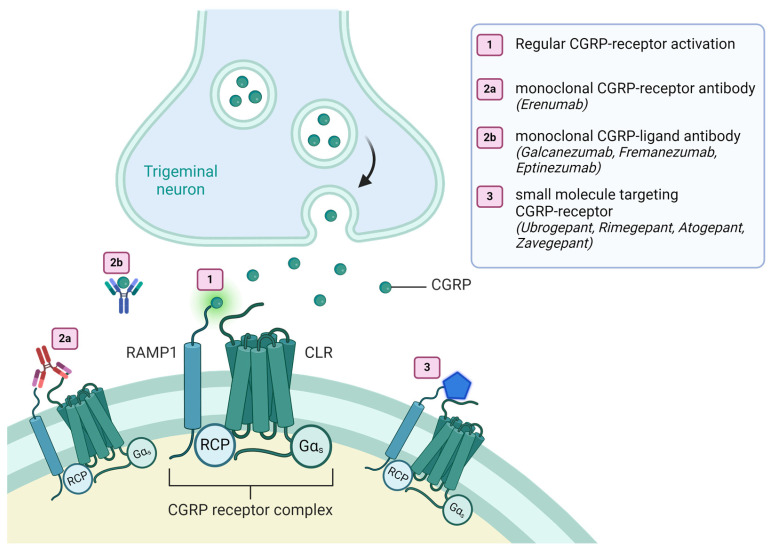
Targets of CGRP-targeting therapies at the canonical CGRP-receptor. (1) Illustrates the activation of the CGRP receptor complex by its endogenous ligand CGRP. (2a) Shows a mAb-dependent blockage of the CGRP receptor. (2b) Demonstrates a mAb-dependent binding of the CGRP ligand. (3) Shows an inactivation of the CGRP receptor complex due to small molecules (gepants). CGRP (Calcitonin-gene related peptide), CLR (Calcitonin receptor-like receptor), Gα_S_ (Gs alpha subunit), RAMP1 (Receptor activity modifying protein 1), RCP (Receptor Component Protein).

**Figure 2 ijms-25-11685-f002:**
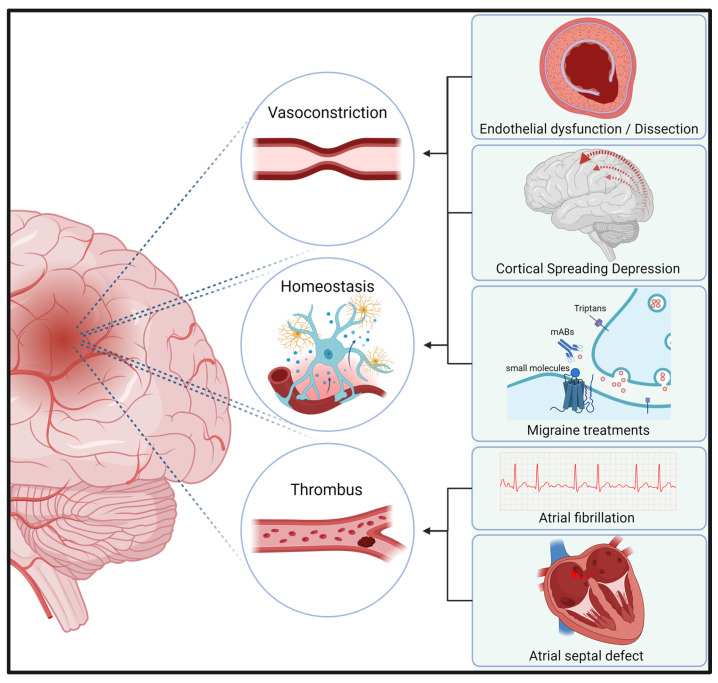
Risk factors for stroke in migraineurs. This figure illustrates the most important studied risk factors for stroke in migraineurs. mABs (monoclonal Antibodies).

**Table 1 ijms-25-11685-t001:** Roles of CGRP in stroke. Bcl-2 (B-cell lymphoma 2), BBB (blood-brain barrier), CBF (cerebral blood flow), CREB (cyclic AMP response element-binding protein), CGRP (calcitonin gene-related peptide), MAPK (mitogen-activated protein kinase), ERK (extracellular signal-regulated kinase), JNK (c-Jun N-terminal kinase), p38 (p38 mitogen-activated protein kinase).

Role of CGRP in Stroke	Molecular Mechanism	Effect in Stroke	Comments
Blood-Brain Barrier (BBB) Influence [10,54,55]	CGRP influences BBB permeability	Increases permeability, which can lead to cerebral edema; conflicting findings on protective effects against BBB injury	Increased permeability may be linked to both beneficial or detrimental outcomes depending on conditions
Vasodilation [56,57]	CGRP and adrenomedullin act as vasodilators, especially in microvasculature	Maintains cerebral blood flow (CBF) during ischemic conditions; counteracts hypoperfusion	Helps in regulating blood flow, particularly during systemic hypotension
Antioxidative Effects [58,59]	CGRP reduces oxidative stress during reperfusion injury	Protects against oxidative stress induced by reintroduction of blood flow post-ischemia	Beneficial in mitigating damage from reperfusion injury
Anti-Inflammatory Effects [9,60,63,64]	CGRP (and adrenomedullin) show anti-inflammatory properties	Reduced inflammation	Supports neuronal recovery by attenuating inflammatory response
Neuroprotection via CREB Pathway [8,61,62]	Activates CREB pathway leading to increased Bcl-2, reduction of caspase-3	Anti-apoptotic effect, protecting neurons	Stabilizes mitochondrial membrane via Bcl-2/BAX pathway
Modulation of MAPK Pathways [63]	Increases ERK phosphorylation, reduces JNK and p38 phosphorylation	Improves neuronal function, reduces apoptosis	MAPK pathway modulation contributes significantly to neuroprotective role

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
