# Peer review of "Novel Calcitonin Gene-Related Peptide (CGRP) Interfering Migraine Therapies and Stroke—A Review"

_ijms, 2024, doi:10.3390/ijms252111685_

Round 1

Reviewer 1 Report

Comments and Suggestions for Authors

My report for ijms-3272210, a review about anti-CGRP(r) mAbs & gepants for migraine (acute and preventive) therapy and stroke :

ms-3272210 is a review manuscript about the new anti-CGRP(r) therapies for migraine and their possible relation with stroke. It examines the physiological effects of GGRP on the vascular system  and stroke (vasodilation, levels in acute stroke and in other vascular diseases), the risk factors for stroke in migraineurs and also the possible effect of the new anti- migraine treatment (acute and preventive) anti-CGRP mAbs and gepants on the vascular system and on stroke (Cardiovascular risk in clinical trials). It highlights the potential risk in subgroups of migraine patients (with comorbidities and older age) of  cerebrovascular AE with the new therapies. For sure there is a need for further research. The review is well written, interesting for your readers, helpful for headache specialists and of scientific soundness. There are not any major or minor issues.

Please accept in current form.

Thank you for submitting your interesting and helpfull manuscript. 

Author Response

Dear Reviewer 1,

Thank you for your positive and encouraging feedback on manuscript IJMS-3272210. We are pleased to hear that you found the review both well-written and scientifically sound, and that its exploration of anti-CGRP therapies’ potential cerebrovascular effects is relevant and beneficial for specialists.

We appreciate your recommendation to accept the manuscript in its current form, and we thank you for your time and thoughtful comments that support the contribution of this review to the field.

Kind regards,

G. Broessner and M. Eller

Reviewer 2 Report

Comments and Suggestions for Authors

I am reviewing the manuscript "Novel Calcitonin Gene-Related Peptide (CGRP) Interfering Migraine Therapies and Stroke – A Review."

This paper is a scoping review that examines the relationship between migraine and stroke, with a specific focus on the potential impact of anti-CGRP therapies on stroke risk. The topic is relevant and holds considerable interest for specialists in headache and stroke, as well as for general neurologists.

The paper is clearly written and well-structured. However, some sentences are brief and would benefit from revision to enhance the fluency and precision of the language.

This review provides valuable insights into the association between anti-CGRP therapies and stroke risk in individuals with migraine and highlights important gaps in current research. With improvements in language and sentence structure, the manuscript has the potential to make a significant contribution to the field.

Author Response

Dear Reviewer 2,

Thank you for your insightful and constructive feedback on the manuscript "Novel Calcitonin Gene-Related Peptide (CGRP) Interfering Migraine Therapies and Stroke – A Review." We appreciate your recognition of the topic’s relevance and the paper's potential contribution to the field. Your acknowledgment of the manuscript's clarity and structure is encouraging, and we are grateful for your suggestions to improve the fluency and precision of the language.

We carefully reviewed and revised the manuscript to enhance sentence structure and language flow, ensuring it increases the readability.

Thank you once again for your thoughtful review and support in refining this work.

Kind regards,

G. Broessner and M. Eller

Reviewer 3 Report

Comments and Suggestions for Authors

This is a comprehensive bibliographic review discussing the role of CGRP in stroke and CV risk in general, as well as the role of anti-CGRP treatments.

The review is well written and informative, covering all aspects of the matter while staying loyal to evidence-based practices.

Some minor comments in order to improve the overall quality of the manuscript are as follows:

Line 41: groth-factor = growth-factor

Line 40-41-42:  Suggesting … syntax error.

Line 63: According to the 2021 GBD (published in 2024) Migraine is the 3rd disease with the most YLDs. Ref 13 is outdated.

Line 64: “CGRP levels are elevated in the jugular vein during an acute migraine attack.” Ref required.

Line 133: …other reviews”  - Refs missing.

Lines 141-142: n a Danish population-based, nationwide study a connection of migraine with aura and cardioembolic stroke due to atrial fibrillation (AF) has been reported” – Ref is missing.

Line 201: “CBF” Abbreviations should be explained in full when reported for the first time in the manuscript

Line 202-203” “CGRP offers both vasodilatory and antioxidative properties. effects.”

Line 278:  “Title 7. Cardiovascular risk signals in non-clinical”  do you mean “studies” ??

Line 237 to 250:  You could additionally refer to this article: https://headachejournal.onlinelibrary.wiley.com/doi/pdf/10.1111/head.14485

Lines 284 to 286 repeats the same as lines 287 to 288.

Since “placebo” and “vehicle” are not interchangeable terms you should use one of them in lines 283, 288, 291, 296, 298.

It would be of great interest to discuss the difference in terms of stroke risk in MA and MO and the pathophysiologic implications of this difference in the CGRP role. As well as the sex differences on the same matter.

I suppose that references into brackets should be placed before the dot throughout the manuscript.   [1].    And not   .[1]  .

In case the very informative figures are original, please add the utility used to create them.  Otherwise state the origin, citation and permission for use.

Author Response

Dear Reviewer 3,

Thank you for your detailed and constructive feedback on our manuscript. We appreciate your recognition of the manuscript’s comprehensive coverage and adherence.

We will address each of your specific comments to enhance the quality and clarity of the manuscript.

Comment 1: Line 41: groth-factor = growth-factor
Response 1: Thank you for pointing out this typo, we changed it accordingly.

Comment 2: Line 40-41-42:  Suggesting … syntax error.
Response 2: We changed the wording to: These findings propose...

Comment 3: Line 63: According to the 2021 GBD (published in 2024) Migraine is the 3rd disease with the most YLDs. Ref 13 is outdated.
Response 3: Thank you for the clarification, we changed it accordingly.

Comment 4: Line 64: “CGRP levels are elevated in the jugular vein during an acute migraine attack.” Ref required.
Response 4: We added the Ref.

Comment 5: Line 133: …other reviews”  - Refs missing.
Response 5: Refs added.

Comment 6: Lines 141-142: n a Danish population-based, nationwide study a connection of migraine with aura and cardioembolic stroke due to atrial fibrillation (AF) has been reported” – Ref is missing.
Response 6: Ref. added.

Comment 7: Line 201: “CBF” Abbreviations should be explained in full when reported for the first time in the manuscript
Response 7: Thank you for having an eye on the detail. We added the full word of the abbreviation.

Comment 8: Line 202-203” “CGRP offers both vasodilatory and antioxidative properties. effects.”
Response 8: We deleted the redundant point.

Comment 9: Line 278:  “Title 7. Cardiovascular risk signals in non-clinical”  do you mean “studies” ??
Response 9: Thank you for pointing out that a word is missing. Yes, we mean studies and added the word accordingly.

Comment 10: Line 237 to 250:  You could additionally refer to this article: https://headachejournal.onlinelibrary.wiley.com/doi/pdf/10.1111/head.14485
Response 10: Thank you for recommending us this relevant reference. We have reviewed the article by Kudrow et al. (2023) and integrated relevant findings into the section "Cardiovascular risk signals in clinical trials using monoclonal antibodies and gepants".

Comment 11: Lines 284 to 286 repeats the same as lines 287 to 288
Response 11: We reviewed the section and agree that lines 284 to 286 are redundant and deleted them. 

Comment 12: Since “placebo” and “vehicle” are not interchangeable terms you should use one of them in lines 283, 288, 291, 296, 298.
Response 12: Thank you for  pointing out this important issue. We changed all terms to "placebo".

Comment 13: It would be of great interest to discuss the difference in terms of stroke risk in MA and MO and the pathophysiologic implications of this difference in the CGRP role. As well as the sex differences on the same matter.
Response 13: Thank you for that idea. We totally agree with you and added the following lines: 138-141, 185-188, 372-379

Comment 14: I suppose that references into brackets should be placed before the dot throughout the manuscript.   [1].    And not   .[1]  .
Response 14: Thank you, we changed it accordingly.

Comment 15: In case the very informative figures are original, please add the utility used to create them.  Otherwise state the origin, citation and permission for use.
Response 15: In fact, these figures are original and were created exclusively for this review by our Co-Author F. Frank. The software he used was Adobe Photoshop Version 26.0, Adobe In Design Version 2024, Adobe Illustrator Version 29.0 and BioRender.

Thank you once again for your insightful suggestions, which will undoubtedly strengthen the manuscript’s contribution to the field.

Kind regards,

G. Broessner and M. Eller